# Regulation of CLK1 Isoform Expression by Alternative Splicing in Activated Human Monocytes Contributes to Activation-Associated TNF Production

**DOI:** 10.3390/cells14231925

**Published:** 2025-12-03

**Authors:** Maurice J. H. van Haaren, Alejandra Bodelón, Lyanne J. P. M. Sijbers, Rianne Scholman, Lucas W. Picavet, Jorg J. A. Calis, Sebastiaan J. Vastert, Jorg van Loosdregt

**Affiliations:** 1Center for Translational Immunology, University Medical Center Utrecht, 3584 CX Utrecht, The Netherlands; 2Division of Pediatric Rheumatology and Immunology, Wilhelmina Children’s Hospital, 3584 CX Utrecht, The Netherlands

**Keywords:** alternative splicing, monocytes, CLK1, long-read RNA-seq

## Abstract

Alternative splicing is a key regulator of immune regulation by enabling rapid and context-specific responses. However, the role of splicing regulators such as CDC-like kinase 1 (CLK1) in monocyte biology remains poorly defined. Here, we identify and characterize distinct CLK1-splice isoforms in human CD14^+^ monocytes using long-read RNA sequencing. In resting monocytes, we observe predominant expression of a truncated isoform lacking exon 4 (CLK1Δ4), which undergoes nonsense-mediated decay resulting in minimal protein output. Lipopolysaccharide (LPS) stimulation induces a shift toward the full-length isoform (CLK1+4), associated with increased transcript stability and protein expression. This splicing switch was confirmed by RT-qPCR, short-read RNA sequencing, and Western blot analysis. Pharmacological inhibition of CLK1 selectively reduced TNFα production without affecting cell viability, implicating that the isoform shift enhances pro-inflammatory signaling. These findings uncover a stimulus-dependent splicing mechanism that modulates monocyte activation through differential CLK1 isoform expression and suggest a potential therapeutic avenue by targeting splicing regulators in immune-related disease with an established role of activated monocytes.

## 1. Introduction

The immune system relies on rapid and precise adaptations in gene expression to mount effective defenses against invading pathogens. A key mechanism enabling its flexibility is alternative splicing, which expands transcriptomic and proteomic diversity and allows for context-specific immune regulation [1,2,3]. In immune cells, alternative splicing orchestrates key aspects of activation, signaling, and cytokine production, and its dysregulation is increasingly recognized as contributor to immune-related diseases [2,4,5,6]. Indeed, splicing quantitative trait loci (sQTLs) have been linked to the susceptibility of immune-mediated diseases in genome-wide association studies (GWAS), implicating splicing variants in conditions such as Systemic Lupus Erythematosus, Rheumatoid Arthritis, and Multiple Sclerosis [7,8].

Among the central regulators of alternative splicing is CDC-like kinase 1 (CLK1), a dual-specificity kinase that can phosphorylate serine/arginine-rich (SR) proteins. CLK1 has been linked to the regulation of viral replication, highlighting its functional relevance in immune regulation [9,10]. These SRSF proteins modulate splice site selection in a position-dependent manner, thereby promoting or preventing exon inclusion depending on whether they bind to exonic or intronic sequences [10]. Additionally, CLK1 has been reported to phosphorylate non-splicing-related substrates including myelin basic protein (MBP) and tyrosine phosphatase (PTP-1B) [11]. Despite the central relevance of splicing in immune cell function, relatively little is known about CLK1 activity and its targets in monocytes. Although specific SRSF proteins have been implicated in monocyte and macrophage function, the upstream regulation of these splicing factors via CLK1 remains poorly characterized [12,13,14,15].

CLK1 itself is subject to post-transcriptional regulation, and given its central role in splicing, may also be regulated by alternative splicing. However, the expression dynamics and functional relevance of CLK1 splice isoforms in human immune cells remain largely unexplored. Previous studies have implicated CLK1 in viral replication (e.g., HIV), but its role in innate immune signaling and inflammation is unknown [9].

In this study, we identify and characterize multiple CLK1 isoforms in primary human CD14^+^ monocytes using long-read RNA sequencing. Upon immune activation with lipopolysaccharide (LPS), we observe a significant shift in isoform usage, favoring the full-length RNA isoforms with increased stability, resulting in increased CLK1 protein expression, and promoting pro-inflammatory cytokine production. These findings provide new insights into monocyte regulation and highlight the broader impact of splicing control in immunology and immune-related diseases.

## 2. Materials and Methods

### 2.1. Cell Culture

Human peripheral blood mononuclear cells (PBMCs) were obtained from healthy donor blood in sodium heparin tubes and isolated using Ficoll-paque plus (Cytiva, Marlborough, MA, USA) density gradient media. Primary human CD14^+^ monocytes were isolated from PBMC using automatic magnetic-activated cell sorting (MACS, Miltenyi Biotech, Bergisch Gladbach, Germany) using human CD14^+^ monocyte isolation kit (Miltenyi Biotec, Bergisch Gladbach, Germany) according to the manufacturers protocol. Ex vivo samples were collected at this stage, to avoid any stimulation from in vitro culture. Primary monocytes were cultured in Roswell Park Memorial Institute (RPMI) 1640 medium (Thermo Fisher, Hampton, NH, USA) supplemented with 10% heat-inactivated fetal calf serum (Sigma-Aldrich, St. Louis, MI, USA), 1% L-glutamine (Life Technologies, Carlsbad, CA, USA), 100 U/mL of penicillin and 100 mg/mL of streptomycin (Gibco, New York, NY, USA). Cells were cultured at 37 °C in 5% CO_2_ and stimulated with 100 ng/mL LPS (Sigma-Aldrich) for 3 h in culture medium.

### 2.2. RNA Extraction and Sequencing: Short-Read and Oxford Nanopore Long-Read RNA-Seq

Monocytes were lysed in TRIzol LS reagent (Thermo Fisher) directly or after LPS activation. RNA was isolated using the Dynabead mRNA Purification Kit (Thermo Fisher) according to the manufacturer’s protocol. For short-read RNA-Seq, the concentration of RNA was quantified using a Qubit RNA HS assay and Qubit fluorometer (Thermo Fisher). Polyadenylated messenger RNA was isolated using Poly (A) beads (NEXTflex), and sequencing libraries were produced using the Rapid Directional RNA-seq kit (NEXTflex). Libraries were sequenced at the Utrecht Sequencing Facility using the Nextseq500 platform (Illumina, San Diego, CA, USA), which produced single end reads of 75 bp. For Native RNA-Seq, 1 µM mRNA per sample was loaded on MinION flowcells and ran for approximately two hours at the Utrecht Sequencing Facility. Raw data was interpreted by Nanopore Technologies sequencing base calling software (*Guppy v3.2.10*) and reads that passed minimum quality were kept for the analysis.

### 2.3. Short-Read RNA-Seq Pipeline

Reads were trimmed using *Trim Galore* v0.6.10 with default parameters and aligned to the reference genome (GENCODE: Release v46 (GRCh38.p14), primary assembly) using *STAR* v2.7.11b [16] in two-pass mode basic. CLK1 uniquely mapped read counts were extracted for the whole gene, only for exon 4 and for 3–4 and 3–5 exon junctions, using a custom script in python and the HTSeq 2.0.5 [17] package.

### 2.4. Oxford Nanopore Long-Read RNA Pipeline

*Flair* v2.0.0 [18] was used to identify a confident full-length isoform-level reference transcriptome in monocytes. Specifically, flair align was used to align long reads to the reference genome (GENCODE: Release v46 (GRCh38.p14), primary assembly) using *minimap2* v2.24 [19] with long-read-specific parameters and—*nvrna* (recommended for native-RNA). *flair correct* with—*nvrna* parameter was used to correct misaligned splice sites using the reference genome annotations and short-read splice junctions. Short-read splice junctions were extracted from the *STAR* junction output of our short-read RNA-Seq dataset, with a custom script that keeps strand-specific information. *FLAIR collapse* was used to collapse corrected reads of all samples in high-confidence isoforms using the criteria: transcripts that have at least 3 full-length supporting reads (80% coverage and spanning 25 bp of the first and last exons) in total with the—*stringent* option, transcripts whose 5′ end was located within 100 bp from the TSS annotated by refTSS [20] and/or TSSclassifier (“relaxed” or “strict”), which are based on the FANTOM CAGE (Cap Analysis of Gene Expression) peak [21] with the *promoters* option, enforcing a 4 out of 6 bp coverage around each splice site and no insertions greater than 3 bp with the *check-splice* option, and keeping isoforms that are a proper subset of another isoform with the *filter comprehensive* option.

*SQANTI3* v5.2.1 [22], including annotated TSS [21] and poly (A) sites, was used to assess the quality of our novel transcriptome. *SQANTI3* was also used to filter and rescue (with default filtering, except RT-Switching) to create the final novel transcriptome. *Flair* v2.0.032 *quantify* was used to extract *tpm* values per isoforms. SUPPA [23] was used to detect splicing events (*generateEvents*) and to perform the differential splicing event analysis (*psiPerEvent* and *diffSplice*) using tpm values. IsoformSwitchAnalyzeR [24] was used for isoform visualization, including predicted protein domains [25] and protein topology predictions [26].

### 2.5. Analyzing CLK1 RNA Expression Using Real-Time PCR

Cells were lysed in RLT buffer (RNeasy kit, Qiagen, Waltham, MA, USA, 74106), and RNA was isolated using the manufacturer’s protocol. cDNA synthesis was performed using an iScript cDNA synthesis Kit (Bio-Rad, South Granville, Australia, 1708890). qPCR reaction was performed with SYBR Select master mix (Thermo Fisher Scientific, Waltham, MA, USA, 13266519) and the primer pair listed in Appendix A. The average expression of both housekeeping genes *B2M* and *RPL13A* was used to ensure accurate normalization; in all experiments their levels did not change significantly upon LPS stimulation.

### 2.6. Analyzing CLK1 Protein Expression Using Western Blot

Cells were lysed in Laemmli buffer (0.12 M Tris-HCl, pH 6.8, 4% SDS, 20% glycerol, 0.05 µg/µL of bromophenol blue, and 35 mM β-mercaptoethanol). Samples were separated using SDS-PAGE on 12% gel and transferred to a polyvinylidene difluoride (PVDF) membrane (Merck & Co., Rahway, NJ, USA). After blocking with 5% milk powder in 1% tris-buffered saline and Tween 20 (TBST), the membrane was probed with the antibodies indicated in Appendix A and analyzed using enhanced chemiluminescence (Thermo Fisher Scientific, Waltham, MA, USA).

### 2.7. Inhibitors Used to Study RNA Dynamics and Functional Effects

Several chemical inhibitors were used to study RNA dynamics. Actinomycin D (ActD, Sigma-Aldrich, St. Louis, MO, USA) was dissolved in DMSO and used at a final concentration of 10 µg/mL to inhibit transcription. Similarly, cycloheximide (CHX, Merck KGaA, Darmstadt, Germany) was used at 0.5 mg/mL to inhibit translation. CLK1-IN-1 (CLKi, MedChemExpress, Monmouth Junction, NJ, USA) is used to inhibit CLK at 100 µM to detect functional effects. Functional effects were measured using flow cytometry.

### 2.8. Flow Cytometry

Cells were washed in FACS buffer (PBS with 2% FBS, and 0.1% NaN3), fixated and permeabilized with a fixation/permeabilization solution kit (BD) and stained with antibodies [27]. Measurements were performed using the BD FACSCanto™ II flow cytometer, and FlowJo v.10 was used for data analysis.

### 2.9. Statistics

Statistical analysis was performed using Graphpad Prism 10.1.2. The statistical tests used to test significance are specified in the figure legends.

## 3. Results

### 3.1. Identification and Characterization of CLK1 Isoforms in Human Monocytes

To investigate alternative splicing dynamics in monocytes, we performed isoform-specific RNA sequencing analyses. Initially, we used Oxford Nanopore Technologies (ONT) long-read RNA sequencing to profile transcript isoforms from native RNA extracted from human CD14^+^ monocytes, both directly ex vivo (unstimulated) and after a three-hour activation with lipopolysaccharide (LPS). We then used Full-Length Alternative Isoform Analysis of RNA (FLAIR) to detect known and novel expressed isoforms between the two conditions to identify candidate genes showing differential splicing. Among these candidates, CLK1 stood out because of its pronounced isoform changes and its established role in phosphorylating serine/arginine-rich splicing factors (SRSFs), prompting us to investigate its regulation in greater detail.

Ex vivo, so analysis directly after isolation, the predominant CLK1 isoform was a truncated variant lacking exon 4 (CLK1Δ4), which introduces a premature stop codon (PTC). Although CLK1Δ4 has been described previously, its expression in monocytes or within an immunological context has not been described [18,19]. Differential splicing analysis using SUPPA revealed a significant increase in exon 4 inclusion following LPS stimulation (Figure 1A). We categorized CLK1 transcripts into three groups: those including exon 4, those excluding exon 4, and those isoforms containing intron retention events (Figure 1B). The shift in CLK1 isoform usage was further visualized with sashimi plots using Integrative Genomics Viewer (Figure 1C) [28]. We detected a clear shift in relative isoform usage towards a marked increase in exon 4-including CLK1 transcripts after activation (Figure 1D).

Although average CLK1 expression levels remained relatively similar, expression of the full-length CLK1+4 isoform significantly increased following LPS stimulation, whereas the expression of CLK1Δ4 is significantly reduced (Figure 1E). To validate these findings, we specifically assessed the expression of the CLK1 isoforms in two independent short-read RNA sequencing datasets of human monocytes (our own, and publicly available dataset GSE103501). Again, a relative increase in expression (approximately 30-fold) of CLK1+4 compared to CLK1Δ4 was observed (Figure 1F). These results indicate that monocyte activation induces an isoform switch favoring full-length CLK1 expression.

### 3.2. Validation of CLK1 Isoform Shift at RNA and Protein Levels Reveals Alternative Splicing as a Regulator of CLK1+4 Protein Expression

To confirm the presence and regulation of CLK1 isoforms, RT-qPCR was performed using isoform-specific primers designed to quantify total CLK1, CLK1+4, and CLK1Δ4 transcripts (Figure 2A). Similarly to the RNA sequencing results in Figure 1, LPS stimulation significantly increased both total CLK1 expression and the absolute levels of the CLK1+4 isoform, while the ratio of CLK1+4 relative to CLK1Δ4 also rose (Figure 2B). Moreover, RNA-seq analysis demonstrated that CLK1 is expressed at significantly higher levels than the other three CLK paralogs in ex vivo monocytes (one-way ANOVA; Appendix A). Together, these data indicate that the observed isoform shift occurs post-transcriptionally during monocyte activation.

To assess whether this isoform shift is reflected at the protein level, Western blot analysis was conducted on CD14^+^ monocytes isolated from PBMCs, with and without LPS stimulation. A clear increase in CLK1 protein levels was observed, consistent with increased CLK1+4 transcript abundance (Figure 3A,B). CLK1Δ4 protein was undetectable, suggesting that this isoform transcript is not translated and/or is rapidly degraded, possibly via nonsense-mediated decay (NMD). As the antibody was raised against an epitope within amino acids 1–130, and exon 4 lies outside this region, we deem it unlikely that the antibody is not able to recognize the CLK1Δ4 protein.

Collectively, these findings indicate that exon 4 inclusion governs CLK1 protein expression in activated monocytes, revealing alternative splicing as a regulator of CLK1-mediated signalling.

### 3.3. CLK1 Isoforms Exhibit Differential Stability and Nonsense-Mediated Decay Susceptibility

To examine the post-transcriptional fate of CLK1 isoforms, we assessed isoform-specific transcript stability using Actinomycin D (ActD) to inhibit transcription. CLK1Δ4 transcripts were rapidly degraded, while CLK1+4 transcripts remained relatively stable (Figure 4A). To further explore the contribution of NMD, translation was blocked using Cycloheximide (CHX), thereby preventing the recognition of premature termination codons and initiation of NMD. CHX treatment led to a marked accumulation of CLK1Δ4 transcripts but did not affect CLK1+4 (Figure 4B), suggesting that CLK1Δ4 is actively degraded via the NMD pathway. These data confirm that CLK1+4 is a stable, protein-coding isoform, while CLK1Δ4 is post-transcriptionally silenced via NMD.

### 3.4. CLK1 Activity Promotes TNF Production in Stimulated Monocytes

Given the stimulus-induced increase in CLK1+4 and its regulation at the protein level, we hypothesized that CLK1 activity contributes to monocyte effector function. To investigate the role of CLK1 in monocyte activation, human CD14^+^ monocytes were stimulated with LPS in the presence or absence of the CLK1 inhibitor CLK1-IN-1, and TNF expression was determined as measure of monocyte activation. While no significant effects on cell viability were observed as a result of CLK1-IN-1 treatment, CLK1 inhibition resulted in a significant reduction in TNFα protein levels (Figure 5A–D). These findings indicate that CLK1 activity promotes monocyte activation and suggest that the LPS-induced isoform shift toward CLK1+4 supports pro-inflammatory cytokine production.

## 4. Discussion

Our study identifies a previously unrecognized mechanism of post-transcriptional regulation during monocyte activation, driven by alternative splicing of the CLK1 gene. We characterized a dynamic shift from a truncated isoform skipping exon 4 (CLK1Δ4) to the full-length CLK1+4 isoform upon LPS stimulation. CLK1Δ4 transcripts were less stable due to nonsense-mediated decay (NMD), whereas CLK1+4 is stable and translated into functional protein. These findings support a model in which CLK1Δ4 acts as a regulatory checkpoint in resting monocytes, limiting kinase availability and maintaining immune quiescence, while selective degradation of CLK1Δ4 and increased exon 4 inclusion upon stimulation facilitate a rapid, isoform-specific increase in CLK1 activity.

While our study establishes the CLK1Δ4-to-CLK1+4 splicing switch during monocyte activation, the upstream mechanisms driving LPS-induced exon 4 inclusion remain to be determined. Prior work on CLK1 autoregulation has shown that exon 4 usage can be controlled by a defined set of SR and SR-related proteins: TRA2β, TRA2α, SRSF4, SRSF5, SRSF7, SRSF8, and SRSF9 promote exon 4 inclusion, whereas SRSF3, SRSF10, and SRSF12 favor exon 4 skipping [29,30]. These studies demonstrate that CLK1 splicing is highly sensitive to the balance of SR-protein activities and that exon 4 serves as a regulatory hub coordinating productive versus non-productive isoform generation. In the context of monocyte activation, it is therefore plausible that LPS-driven signaling alters the activity, or expression of one or more of these factors, either directly via TLR4-NF-κB/MAPK pathways or indirectly through changes in SR-protein kinase activity, including CLK1 itself. Such a mechanism would provide a coherent model in which LPS stimulation enhances the activity of exon-4-inclusion factors (e.g., TRA2β or SRSF7) or suppresses exon-skipping factors (e.g., SRSF3), thereby shifting the balance toward productive CLK1+4 isoform formation. This model aligns with an autoregulatory feedback loop described for CLK1, in which increased production of the full-length isoform amplifies SR-protein phosphorylation, stabilizing exon 4 inclusion and coordinating broader splicing programs. Although our study does not directly identify which factors mediate the LPS-dependent shift, integrating these mechanistic insights highlights testable candidates and provides a rationale for future experiments using isoform-specific reporters, targeted knockdown/overexpression of individual SR proteins, or long-read RNA sequencing to map the regulatory cascade controlling exon 4 selection during monocyte activation.

As a key modulator of pre-mRNA splicing through SR-protein phosphorylation, CLK1 isoform composition can influence numerous transcripts. Changes in the balance between productive and nonproductive isoforms alter cellular CLK1 dosage, fine-tuning splice-site selection in a context-dependent manner. CLK1-dependent SR-protein regulation also affects mRNA export, stability, and transcription–splicing coupling, providing multiple routes through which isoform remodeling could impact inflammatory gene expression. Although we did not directly assess global splicing changes, this framework offers a plausible explanation for how altered CLK1 isoform expression contributes to the TNF-associated phenotype observed here. Functionally, inhibition of CLK1 activity using CLK1-IN-1 significantly reduced TNFα expression following LPS stimulation, underscoring a role for CLK1 in driving pro-inflammatory signaling. While CLK1-IN-1 exhibits strong inhibitory activity toward CLK1, its cross-inhibition of CLK2–4 means that some downstream effects may reflect off-target kinase inhibition, thereby limiting isoform-specific interpretations. Future studies employing inducible knockout models or isoform-specific degrons, combined with long-read RNA sequencing, could further resolve the distinct contributions of CLK family members across immune contexts [31].

In principle, overexpression of full-length CLK1 in primary monocytes would be a valuable approach to further validate the proposed model linking increased CLK1 availability to enhanced TNFα production. However, such experiments proved technically challenging and difficult to interpret in our hands. Primary human monocytes exhibit a strong activation response upon isolation and culture, driven by adhesion to plastic surfaces and exposure to culture medium. Even without stimulation, this handling-induced activation rapidly induces a broad transcriptional program, including IL8, CCL2, CCL7, IL7R, STAT4, IL1R1, IL10, TNF, and CLK1, within only a few hours. The introduction of exogenous nucleic acids further amplifies this activation state, resulting in a plateau beyond which additional stimuli, including LPS, fail to elicit measurable increases in TNFα. Future work employing delivery methods with reduced immunogenicity or monocyte-like model systems with lower basal responsiveness may make such validation feasible.

Although CLK1Δ4 is likely non-functional due to its instability and lack of detectable protein, its regulated production may serve a protective role, preventing premature or inappropriate CLK1 activity. Dysregulation of this splicing switch, through mutation, impaired exon recognition, or defective RNA surveillance, could lead to aberrant retention of CLK1Δ4 or loss of CLK1+4, thereby disrupting broader splicing dynamics or immune responsiveness [32,33,34,35]. Such defects may underlie immune dysfunction in chronic inflammatory or autoimmune disorders.

Similar splicing-based control mechanisms have been described for other immune genes. For example, stimulus-induced isoform switching in CD45, IL7R, and FOXP3 modulates signaling thresholds, receptor activity, and T cell function [32,36,37]. These parallels support the broader concept of stimulus-dependent isoform switching in immune regulation. In this context, the regulated isoform switch in CLK1 may represent not just a gene-specific mechanism but a general paradigm for post-transcriptional control of immune function. Whether CLK1 splicing is part of a wider network of co-regulated events or uniquely adapted to monocytes remains to be determined. Perturbations in this switch, such as failure to degrade non-productive isoforms or premature expression of active variants, may lower the activation threshold of immune cells and contribute to disease pathogenesis in conditions such as rheumatoid arthritis and cancer [13,38,39,40,41,42,43,44]. However, these hypotheses require further experimental validation.

Although we modulated CLK1 activity pharmacologically, targeting the splicing event itself may offer a more selective therapeutic strategy. Splice-switching oligonucleotides (SSOs) designed to promote exon 4 exclusion could increase CLK1Δ4 expression and dampen pro-inflammatory signaling. Nonetheless, this approach warrants caution. Modulating CLK1 splicing may influence global spliceosome function through effects on SR-protein phosphorylation, and while morpholino-based SSOs have been reported not to elicit TLR or related innate immune pathways, other SSO chemistries can activate monocyte innate immunity via TLR7/8/9. [13,38,39,40,41,42]. Perturbations in this switch, either through failure to degrade non-productive isoforms or premature expression of active variants, may lower the activation threshold of immune cells and contribute to disease pathogenesis. Thus, therapeutic strategies targeting CLK1 splicing will require careful optimization to balance efficacy with immunogenicity.

In summary, our findings establish CLK1 isoform switching as a rapid and regulated mechanism that modulates monocyte activation. This work contributes to the growing recognition of alternative splicing as a dynamic layer of immune regulation, with potential implications for the development of isoform-specific immunomodulatory therapeutics [13,38,39,40,41,42].

## 5. Conclusions

Our study identifies alternative splicing of CLK1 as regulatory mechanism controlling monocyte activation. We demonstrate that LPS stimulation induces a switch from the unstable, truncated CLK1Δ4 isoform to the stable, protein-coding CLK1+4 isoform, thereby increasing CLK1 protein levels and promoting pro-inflammatory cytokine production. This isoform-specific regulation highlights how splicing can fine-tune kinase activity and downstream immune responses in a rapid and context-dependent manner. Beyond CLK1, these findings illustrate a broader principle of stimulus-dependent isoform switching as a dynamic layer of immune regulation, with potential implications for autoimmune and inflammatory diseases. Finally, our results provide a foundation for exploring isoform-specific therapeutic strategies targeting CLK1 or other splicing-regulated immune modulators, while underscoring the need to balance efficacy with potential effects on global splicing networks.

## Figures and Tables

**Figure 1 cells-14-01925-f001:**
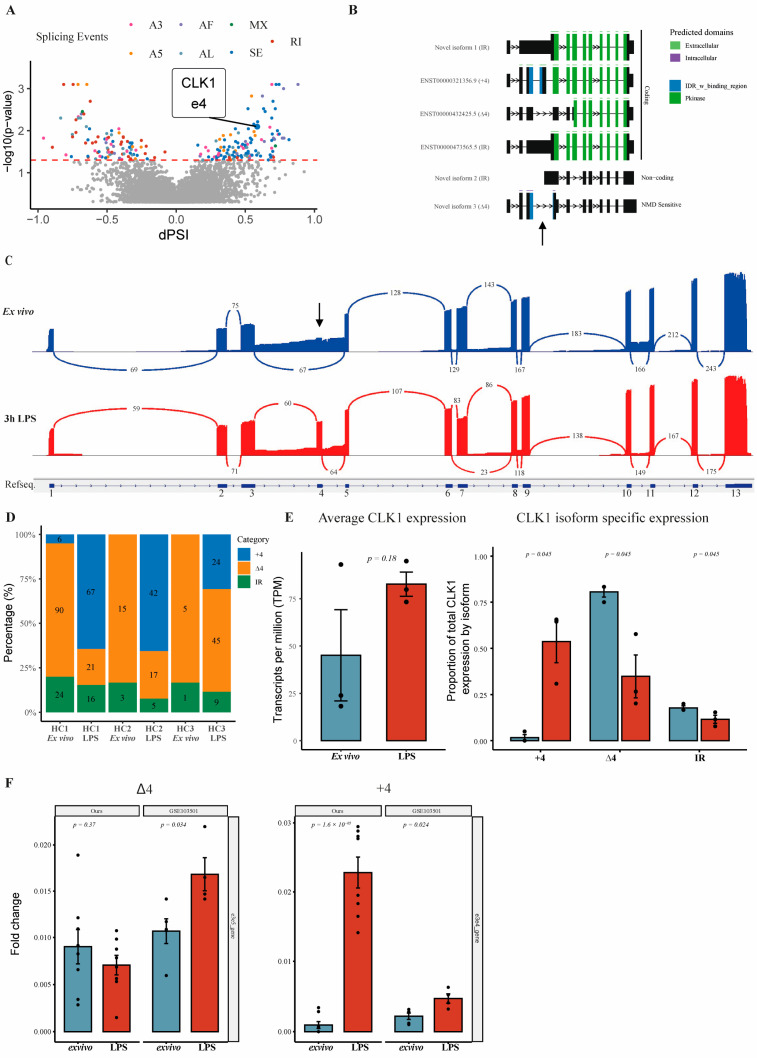
Detection of *CLK1* Isoform Shift in Monocytes Using Oxford Nanopore Long-Read Sequencing. (**A**) Volcano plot showing differentially specific splicing events between LPS and ex vivo using SUPPA. (**B**) Isoforms detected by FLAIR, plotted using IsoformSwitchAnalyzeR [24]. (**C**) Sashimi plots from third-generation RNA sequencing. (**D**) Summary graph indicating percentual presence of exon 4 inclusion (+4), exclusion (Δ4), or intron retention (IR) from the total expression. (**E**) Average CLK1 gene expression detected with Nanopore and isoform specific proportions. Statistical significance was determined using paired *t* test. (**F**) Validation of the splicing shift in healthy donors using two next-generation sequencing datasets: exon 4 reads using either 3–4 and 3–5 junction reads normalized by total gene expression in Reads per Million. Statistical significance was determined using unpaired *t* test.

**Figure 2 cells-14-01925-f002:**
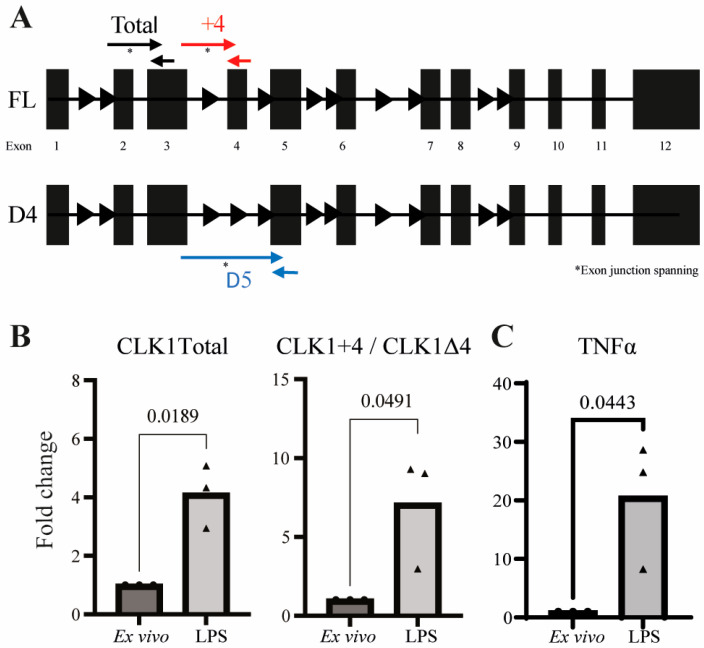
Validation of *CLK1* expression on RNA Level using RT-qPCR. RT-qPCR analysis was performed on total *CLK1* and specific isoforms *CLK1Δ4* and *CLK1+4*, using primers as schematically represented in (**A**). Expression levels were normalized to total expressed *CLK1*, and fold changes were calculated relative to ex vivo samples, using *B2M* and *RPL13A* as housekeeping genes (**B**). Statistical significance was determined using an unpaired *t*-test (*n* = 3). Expression of TNFα was quantified as an indicator of immunological activation and calculated relative to ex vivo samples, using B2M and RPL13A as housekeeping genes (**C**).

**Figure 3 cells-14-01925-f003:**
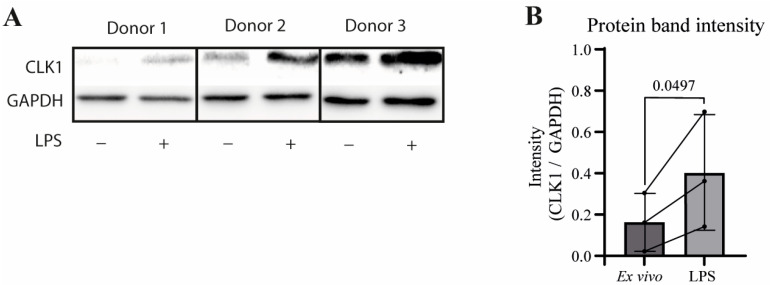
Validation of CLK1 expression on protein level. Western blot analysis was performed to assess CLK1 protein expression following 3 h LPS stimulation (**A**). Quantification of protein levels normalized to GAPDH across three donors showed a consistent increase upon stimulation, with connecting lines indicating paired samples (**B**). Statistical significance was determined using a paired *t*-test (*n* = 3).

**Figure 4 cells-14-01925-f004:**
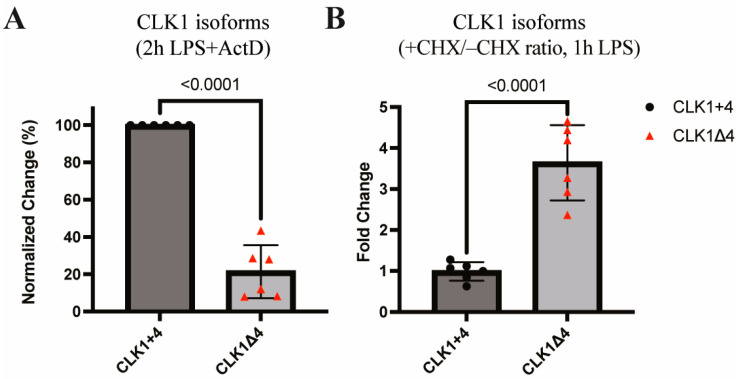
Transcription and translation inhibition indicates that CLK1Δ4 RNA isoform is unstable. RT-qPCR analysis was performed on CLK1 isoforms in the presence of the transcription inhibitor ActD (**A**) or an inhibitor of NMD CHX (**B**). Data are presented as mean ± SEM from six biological replicates. Statistical significance was determined using an unpaired two-tailed *t*-test, with *p*-values < 0.05 considered significant.

**Figure 5 cells-14-01925-f005:**
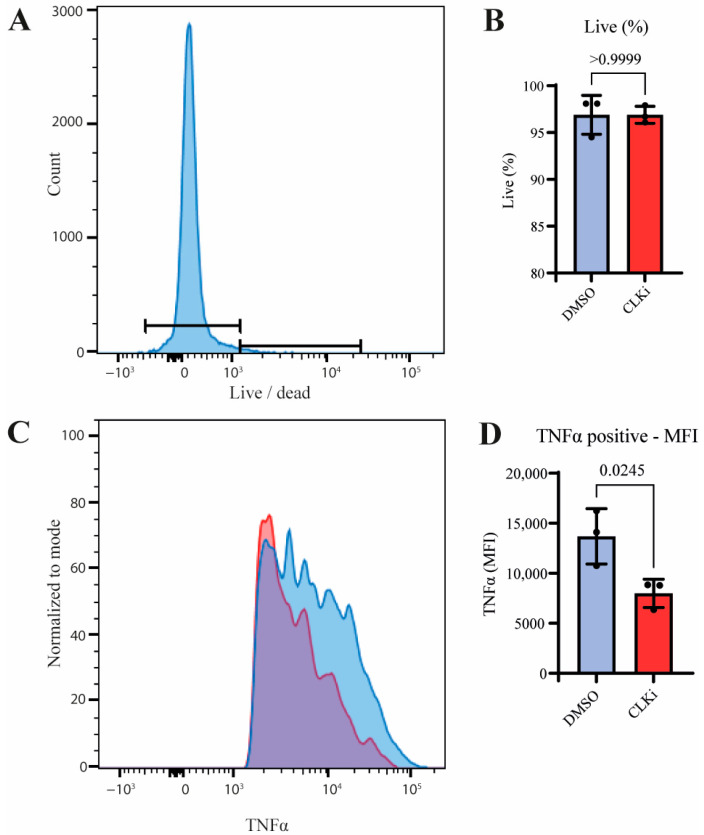
CLK inhibition leads to decreased TNF signaling in stimulated monocytes. Flow cytometry analysis of TNFα expression in monocytes stimulated with LPS in the presence (red) or absence (blue) of 100 µM CLK inhibitor (CLK1-IN-1, CLKi) showed a reduction in TNFα positivity, as illustrated by a representative histogram from one donor (**A**) and summarized for three donors (**C**). Live/dead analysis across the same donors demonstrated no significant differences in viability between treatments (**B**). Quantification of TNFα levels confirmed the reduction upon CLK inhibition (**D**). Statistical significance was determined using a paired *t*-test (*n* = 3).

## Data Availability

All data generated or analyzed during this study are included in this article and Appendix A.

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
