# Peer review of "Regulation of CLK1 Isoform Expression by Alternative Splicing in Activated Human Monocytes Contributes to Activation-Associated TNF Production"

_cells, 2025, doi:10.3390/cells14231925_

Round 1
Reviewer 1 Report
Comments and Suggestions for Authors
Comments on van Haaren et al., “Regulation of CLK1 Isoform Expression by Alternative Splicing in Activated Human Monocytes Contributes to Activation-Associated TNF Production”
This manuscript describes an alternative splicing isoform switch in the mRNA encoding CLK1 in activated vs. resting CD14+ monocytes. Although this splicing change has been documented before, it has not been investigated specifically in resting and LPS activated monocytes, and the results have implications in the regulation of production of inflammatory cytokines.
The manuscript is generally well written and presented, although I think that some increased discussion on several points below, and one potentially simple additional experiment would add robustness to the paper.
The proposed ‘pathway’ suggests that production of full length CDK1protein ultimately triggers production of TNFa. Although the authors show that the converse is true, i.e., that reduction of the amount of full length CDK1 mRNA correlates with reduction of TNFa, it would be good to show that increase expression of CDK1 in unstimulated and stimulated cells results in increased TNFa expression. This could be accomplished by transfection of these monocytes with mRNA (or DNA) encoding full length (and D4, as a control). Fairly efficient protocols for transfection of monocytes from PBMCs have been established (e.g., Moradian, SciRep 2020).
Although the details of the pathway between CLK1 and TNFa expression are not specified, since CLK1 affects the expression/activity of SFs, it would be good to show whether or not the mRNA for TNFa is affected by the state of splicing/expression of CLK1.
The authors should examine the SF recognition sites near exon 4 of CLK1 to suggest which SFs are activated or silenced in order to trigger inclusion of exon 4 during stimulation.
Minor points:
Line 141: Units for the CLK inhibitor are odd. Concentration should be either mM or mmol/ml, not mM/ml.
It should be emphasized that all kinase inhibitors have off target effects on other kinases. Although the authors point to cross inhibition of the other CLKs, it is important to keep in mind that some downstream effects in kinase inhibition studies may be due in part to these off target effects.
Figure 1D: in the inset of this panel, the blue is labeled “-4”, but this should be “+4”.
Line 209: should state specifically if the antibody used to detect CLK1 would also recognize CLK1D4, i.e., does the hypothetical truncated protein contain the epitope that CLK1 has (or is the epitope missing because the protein is truncated). The description of the antibody in the table is insufficient to know this.
Figure 4A: the label on the y-axis is “Normalized Fold Change (%)”. This should be either “Normalized change (%)” or “Normalized Fold Change”.
Line 299: it is notable that some SSOs that are constructed from morpholino oligonucleotides are reported not to trigger TLR or other immune recognition pathways.
Author Response
This manuscript describes an alternative splicing isoform switch in the mRNA encoding CLK1 in activated vs. resting CD14+ monocytes. Although this splicing change has been documented before, it has not been investigated specifically in resting and LPS activated monocytes, and the results have implications in the regulation of production of inflammatory cytokines.
The manuscript is generally well written and presented, although I think that some increased discussion on several points below, and one potentially simple additional experiment would add robustness to the paper.
The proposed ‘pathway’ suggests that production of full length CDK1protein ultimately triggers production of TNFa. Although the authors show that the converse is true, i.e., that reduction of the amount of full length CDK1 mRNA correlates with reduction of TNFa, it would be good to show that increase expression of CDK1 in unstimulated and stimulated cells results in increased TNFa expression. This could be accomplished by transfection of these monocytes with mRNA (or DNA) encoding full length (and D4, as a control). Fairly efficient protocols for transfection of monocytes from PBMCs have been established (e.g., Moradian, SciRep 2020).
Response 1:
Thank you for this suggestion. This would indeed be a valuable validation experiment and is a point we had already considered during an earlier phase of the project. However, there is a technical limitation that renders these experiments highly challenging and, in our hands, unreliable.
We observed that the mere act of isolating and culturing primary monocytes is sufficient to induce a markedly increased activation profile. This activation is triggered by their interaction with (culture) plastic surfaces and culture medium. In fact, as early as 4 hours after isolation from blood, simply by culturing monocytes in medium without any stimulation, we detect a significant upregulation of numerous activation-associated genes, including IL8, CCL2, CCL7, IL7R, STAT4, IL1R1, IL10, and many others. Similarly, culturing alone significantly increases the expression of both TNF and CLK1.
Moreover, the introduction of foreign DNA/RNA through transfection further amplifies this activation state, which rapidly reaches a plateau. At this point, any additional activation, for example through increased CLK1 expression, can no longer be reliably measured. This is consistent with our previous transfection experiments, where even the addition of LPS, a hallmark monocyte activator, failed to further enhance TNF expression in transfected cells.
As this is a highly relevant point, we have now addressed and discussed this limitation in the Discussion section of the manuscript.
Although the details of the pathway between CLK1 and TNFa expression are not specified, since CLK1 affects the expression/activity of SFs, it would be good to show whether or not the mRNA for TNFa is affected by the state of splicing/expression of CLK1.
Response 2: Thank you for your suggestion, we have added RNA expression for TNF in figure 2, to indicate a direct correlation between TNF upregulation and increased CLK+4 expression.
The authors should examine the SF recognition sites near exon 4 of CLK1 to suggest which SFs are involved
Response 3:
Thank you for this thoughtful suggestion. We agree that mapping the splicing factor recognition elements around exon 4 would provide an even deeper mechanistic understanding of the regulation of CLK1 splicing. This question has been explored extensively in previous systematic analyses using mutational mapping and targeted splicing factor perturbations. These studies consistently identify specific members of the SR and TRA2 families as key regulators of exon 4 inclusion or skipping. Rather than duplicating these technically demanding experiments, we now integrate the most relevant insights from this literature into the Discussion section of our manuscript. This allows us to place our own findings in a broader mechanistic context and to highlight the specific splicing factors that are most plausible candidates for mediating the stimulus-dependent regulation of CLK1 in monocytes. Importantly, this also helps define clear directions for future experiments designed to build upon the novel observations we present in this manuscript.
Line 141: Units for the CLK inhibitor are odd. Concentration should be either mM or mmol/ml, not mM/ml.
Response 4: Thank you for pointing this out. It is an erroneous notation. It has been corrected.
It should be emphasized that all kinase inhibitors have off target effects on other kinases. Although the authors point to cross inhibition of the other CLKs, it is important to keep in mind that some downstream effects in kinase inhibition studies may be due in part to these off target effects.
Response 5: Thank you for pointing this out. We agree with the reviewer and as such have now adapted the discussion section to discuss this point.
Figure 1D: in the inset of this panel, the blue is labeled “-4”, but this should be “+4”.
Response 6: Thank you for your suggestion, an accidental typing mistake, it is corrected.
Line 209: should state specifically if the antibody used to detect CLK1 would also recognize CLK1D4, i.e., does the hypothetical truncated protein contain the epitope that CLK1 has (or is the epitope missing because the protein is truncated). The description of the antibody in the table is insufficient to know this
Response 7: Thank you for this suggestion. The epitope is between the first and the 130th amino acid. The fourth exon is just past this region and thus this isoform should be theoretically detectable. We have added this information into the text at line 209.
Figure 4A: the label on the y-axis is “Normalized Fold Change (%)”. This should be either “Normalized change (%)” or “Normalized Fold Change”.
Response 8: We agree and have now adapted the axis to Normalized Change (%) as suggested by the reviewer.
Line 299: it is notable that some SSOs that are constructed from morpholino oligonucleotides are reported not to trigger TLR or other immune recognition pathways.
Response 9: Thank you for this information. Morpholino oligonucleotides have come a long way and it is good to recognize this. We have added this information to the text at the mentioned location.
Reviewer 2 Report
Comments and Suggestions for Authors
The authors examined the expression of different CLK1 isoforms in human monocytes in response to LPS stimulation and identified an interesting mechanism of alternative splicing mediated regulation of expression. The study is well designed and executed with novel discoveries. Thus, the manuscript is fit for publication after minor revisions.
- Since CLK1 itself is a regulator of splicing, does the change of CLK1 isoform expression impact the splicing of other transcripts and how does that relate to the observed TNF phenotype?
Author Response
Since CLK1 itself is a regulator of splicing, does the change of CLK1 isoform expression impact the splicing of other transcripts and how does that relate to the observed TNF phenotype?
Response 1: Thank you for raising this point. To address this important consideration, we have now added a dedicated section in the Discussion section in which we elaborate on the broader implications of CLK splicing.
Reviewer 3 Report
Comments and Suggestions for Authors
The manuscript presents a compelling finding: a stimulus-dependent alternative splicing switch of the CLK1 gene in human monocytes that regulates CLK1 protein expression and contributes to pro-inflammatory cytokine production. The authors clearly identify and validate two main CLK1 isoforms, CLK1Δ4 (truncated) and CLK1+4 (full-length), and demonstrate their differential stability and translation in resting versus activated monocytes. The functional link between CLK1 activity and TNFα production in activated monocytes is an important insight, suggesting a potential therapeutic target in immune-related diseases. The work is technically sound, utilizing advanced techniques like long-read RNA sequencing, RT-qPCR, Western blot, and flow cytometry.
Main Concerns:
1. The study clearly demonstrates that LPS stimulation increases the functional CLK1+4 protein, and inhibiting CLK1 activity reduces TNFα production. However, a definitive, isoform-specific functional link is not established. The authors acknowledge that the CLK1-IN-1 inhibitor is not entirely specific and also affects CLK2-4 activity, which limits the conclusion that CLK1 alone, or specifically the CLK1+4 isoform, drives TNFα production. While the authors suggest future studies using inducible knockout models or isoform-specific degrons, a more immediate improvement would be to discuss or perform an experiment using an inhibitor with greater specificity to CLK1 if one is available, or to at least quantify the relative expression of CLK2, CLK3, and CLK4 in monocytes to estimate their potential contribution to the observed phenotype. A definitive statement on the role of the CLK family should be included.
2. The manuscript convincingly shows that the CLK1Δ4 isoform is subject to NMD because its transcript levels rapidly decrease upon transcription inhibition (ActD) and accumulate upon translation inhibition (CHX). The study establishes the existence of the splicing switch but does not explore the upstream mechanisms regulating the shift from CLK1Δ4 exclusion to CLK1+4 inclusion upon LPS stimulation. Since CLK1 itself is a key regulator of splicing factors (SRSF family), this leaves a critical gap in the understanding of the feedback loop: What splicing factor (or its regulator) is activated by LPS to cause exon 4 inclusion in CLK1? While this may require a separate study, the Discussion should include a more detailed hypothesis about which specific SRSF proteins or other splicing factors are likely involved in mediating the LPS-induced inclusion of exon 4. The authors mention CLK1 phosphorylates SR proteins, suggesting a potential autoregulatory loop or a pathway involving a factor targeted by an LPS signaling cascade.
Minor Concerns:
1. Figure 1E: The y-axis label "Proportion of total expression" in the right-hand graph of Figure 1E is a bit vague. It represents the proportion of total CLK1 expression attributed to each isoform group (+4,Δ4,IR), which should be clearly stated in the figure legend or body text.
2. The schematic in Figure 2A needs to clearly mark the position of Exon 4 (e4) on the FL (Full-Length) and D4 (Δ4) isoforms. This is crucial for understanding the RT-qPCR primer strategy.
3. In Figure 2B, the p-values for the CLK1 Total and CLK1+4/Total comparisons are 0.0075 and 0.0496 respectively. The text accompanying the results should explicitly state that the increase in CLK1+4 is statistically significant (Figure 2B).
4. The Introduction provides a good overview of alternative splicing in immune regulation. However, the transition to CLK1 is somewhat abrupt. Briefly explaining CLK1's known role as a dual-specificity kinase that phosphorylates SRSF proteins and its existing link to viral replication earlier might improve flow.
5. The RT-qPCR results are normalized to B2M and RPL13A as housekeeping genes. It should be confirmed in the Methods section that the expression levels of these housekeeping genes do not change significantly upon LPS stimulation, as is standard practice, to ensure accurate normalization.
Author Response
The study clearly demonstrates that LPS stimulation increases the functional CLK1+4 protein, and inhibiting CLK1 activity reduces TNFα production. However, a definitive, isoform-specific functional link is not established. The authors acknowledge that the CLK1-IN-1 inhibitor is not entirely specific and also affects CLK2-4 activity, which limits the conclusion that CLK1 alone, or specifically the CLK1+4 isoform, drives TNFα production. While the authors suggest future studies using inducible knockout models or isoform-specific degrons, a more immediate improvement would be to discuss or perform an experiment using an inhibitor with greater specificity to CLK1 if one is available, or to at least quantify the relative expression of CLK2, CLK3, and CLK4 in monocytes to estimate their potential contribution to the observed phenotype. A definitive statement on the role of the CLK family should be included.
Response 1: Thank you for this great suggestion. We have looked for inhibitors specific for only CLK1, but these do not exist yet. To further address your point regarding CLK paralogs, we have assessed the expression of all CLK members (CLK1-4) in primary human monocytes (new supplementary figure 3). These data demonstrate that CLK1 is the dominant expressed CLK family member.
The manuscript convincingly shows that the CLK1Δ4 isoform is subject to NMD because its transcript levels rapidly decrease upon transcription inhibition (ActD) and accumulate upon translation inhibition (CHX). The study establishes the existence of the splicing switch but does not explore the upstream mechanisms regulating the shift from CLK1Δ4 exclusion to CLK1+4 inclusion upon LPS stimulation. Since CLK1 itself is a key regulator of splicing factors (SRSF family), this leaves a critical gap in the understanding of the feedback loop: What splicing factor (or its regulator) is activated by LPS to cause exon 4 inclusion in CLK1? While this may require a separate study, the Discussion should include a more detailed hypothesis about which specific SRSF proteins or other splicing factors are likely involved in mediating the LPS-induced inclusion of exon 4. The authors mention CLK1 phosphorylates SR proteins, suggesting a potential autoregulatory loop or a pathway involving a factor targeted by an LPS signaling cascade.
Response 2: Thank you for this thoughtful suggestion, We have added some more depth in the discussion to further explore this topic> more specifically we now discuss two mechanistic studies that used mutational mapping and targeted splicing factor perturbations to identify specific members of the SR and TRA2 families that regulate CLK1 exon 4 inclusion or skipping.We now integrate the most relevant insights from this literature into the Discussion. This allows us to place our own findings in a broader mechanistic context and to highlight the specific splicing factors that are most plausible candidates for mediating the stimulus-dependent regulation of CLK1 in monocytes.
Figure 1E: The y-axis label "Proportion of total expression" in the right-hand graph of Figure 1E is a bit vague. It represents the proportion of total CLK1 expression attributed to each isoform group (+4,Δ4,IR), which should be clearly stated in the figure legend or body text.
Response 3: We agree and have now changed the figure Y axis to “proportion of total CLK1 expression by isoform” which should make more clear.
The schematic in Figure 2A needs to clearly mark the position of Exon 4 (e4) on the FL (Full-Length) and D4 (Δ4) isoforms. This is crucial for understanding the RT-qPCR primer strategy.
Response 4: Thank you for this point. To further elucidate what is described, We have now added an asterisks to indicate primers that are exon junction spanning, additionally we have added numbers to describe the exons.
In Figure 2B, the p-values for the CLK1 Total and CLK1+4/Total comparisons are 0.0075 and 0.0496 respectively. The text accompanying the results should explicitly state that the increase in CLK1+4 is statistically significant (Figure 2B).
Response 5: Thank you for this remark. We agree and have now added the description in the text.
The Introduction provides a good overview of alternative splicing in immune regulation. However, the transition to CLK1 is somewhat abrupt. Briefly explaining CLK1's known role as a dual-specificity kinase that phosphorylates SRSF proteins and its existing link to viral replication earlier might improve flow.
Response 6: Thank you. We have now adapted the introduction to explain this further.
The RT-qPCR results are normalized to B2M and RPL13A as housekeeping genes. It should be confirmed in the Methods section that the expression levels of these housekeeping genes do not change significantly upon LPS stimulation, as is standard practice, to ensure accurate normalization.
Response 7: Thank you for this suggestion. This is very important for accurate normalization, as such, a statement has been added to the method section.
Round 2
Reviewer 1 Report
Comments and Suggestions for Authors
The authors have done a thoughtful and thorough job of addressing prior concerns.
Reviewer 3 Report
Comments and Suggestions for Authors
The authors fairly addressed my previous concerns.